# Use of open-text responses to recode categorical survey data on postpartum contraception use among women in the United States: A mixed-methods inquiry of Pregnancy Risk Assessment Monitoring System data

Nicole K. Richards[1,2,3]*, Christopher P. Morley[2,3], Martha A. Wojtowycz[3,4], Erin Bevec[5,6], Brooke A. Levandowski[3,5]

1 University of Waterloo, Faculty of Applied Health Sciences School of Public Health and Health Systems, Waterloo, Ontario, Canada, 2 SUNY Upstate Medical University, Department of Public Health and Preventive Medicine, New York, New York, United States of America, 3 SUNY Upstate Medical University, Department of Family Medicine Madison-Irving Medical Center, New York, New York, United States of America, 4 SUNY Upstate Medical University, Department of Obstetrics and Gynecology, Syracuse, New York, United States of America, 5 University of Rochester Medical Center, Department of Obstetrics and Gynecology School of Medicine and Dentistry, New York, New York, United States of America, 6 University of Rochester, School of Arts and Science Department of Public Health Sciences, Rochester, New York, United States of America

* nicole.richards@Uwaterloo.ca

## Abstract

### Background

Postpartum contraception prevents unintended pregnancies and short interpregnancy intervals. The Pregnancy Risk Assessment Monitoring System (PRAMS) collects population-based data on postpartum contraception nonuse and reasons for not using postpartum contraception. In addition to quantitative questions, PRAMS collects open-text responses that are typically left unused by secondary quantitative analyses. However, abundant preexisting open-text data can serve as a resource for improving quantitative measurement accuracy and qualitatively uncovering unexpected responses. We used PRAMS survey questions to explore unprompted reasons for not using postpartum contraception and offer insight into the validity of categorical responses.

### Methods and findings

We used 31,208 categorical 2012 PRAMS survey responses from postpartum women in the US to calculate original prevalences of postpartum contraception use and nonuse and reasons for contraception nonuse. A content analysis of open-text responses systematically recoded data to mitigate survey bias and ensure consistency, resulting in adjusted prevalence calculations and identification of other nonuse themes. Recoded contraception nonuse slightly differed from original reports (21.5% versus 19.4%). Both calculations showed

**Data Availability Statement:** The data underlying the results presented in the study are available for request from the CDC PRAMS working group at https://www.cdc.gov/prams/prams-data/researchers.htm.

**Funding:** The author(s) received no specific funding for this work.

**Competing interests:** The authors have declared that no competing interests exist.

that many respondents reporting nonuse may be at a low risk for pregnancy due to factors like tubal ligation or abstinence. Most frequent nonuse reasons were not wanting to use birth control (27.1%) and side effect concerns (25.0%). Other open-text responses showed common themes of infertility, and breastfeeding as contraception. Comparing quantitative and qualitative responses revealed contradicting information, suggesting respondent misinterpretation and confusion surrounding the term "pregnancy prevention." Though this analysis may be limited by manual coding error and researcher biases, we avoided coding exhaustion via 1-hour coding periods and validated reliability through intercoder kappa scores.

## Conclusions

In this study, we observed that respondents reporting contraception nonuse often described other methods of pregnancy prevention and contraception barriers that were not included in categorical response options. Open-text responses shed light on a more comprehensive list of pregnancy prevention methods and nonuse options. Our findings contribute to survey questions that can lead to more accurate depiction of postpartum contraceptive behavior. Additionally, future use of these qualitative methods may be used to improve other health behavior survey development and resulting data.

## Author summary

### Why was this study done?

- Survey respondents can occasionally provide open-ended answers that may be overlooked during quantitative data analysis.

- This study was conducted to study how qualitative inquiry of open-text responses to a survey question related to postpartum contraception use can be used to provide insight into data accuracy within quantitative surveys.

### What did the researchers do and find?

- Using a mixed-method approach, we calculated descriptive statistics from Pregnancy Risk Assessment Monitoring System (PRAMS) survey responses from 31,208 postpartum women in the US who provided categorized reasons for not using contraception and qualitatively analyzed respondents' adjacent open-ended written answers to uncover more detailed information about respondents' postpartum contraception behavior and interpretations of the survey question.

- Through systematic recategorization, we found that qualitative responses added to and altered quantitative contraception behavior reports and that reasons for contraception nonuse are incompletely captured by existing response categories.

**What do these findings mean?**

- These findings suggest that researchers may be able to gain a better understanding of postpartum pregnancy prevention by using mixed-method analyses on categorical and open-ended responses.

- While qualitative inquiry may yield subjective interpretation, rigorous pragmatic data management can strengthen survey questions through open-text responses not captured in existing quantitative data.

- Family planning professionals may be able to obtain significant and detailed information about contraceptive behavior from open-ended responses, and this information may be useful to adjust clinical approaches to postpartum contraception conversations.

## Introduction

Postpartum contraception uptake mitigates unintended pregnancies and inadequate birth spacing, playing a large role in preventing poor birth outcomes and promoting healthy mothers, infants, and families [1,2]. Many factors influence contraception decisions and access, and, therefore, not all people that want to prevent pregnancy use a contraception method [3]. Some primary care clinicians report assumptions that the patient will initiate contraception conversations, and both patients and providers report struggling with contraception counseling due to lack of contraception knowledge and comfort [4,5]. Further, health literacy and numeracy can impact patients' reproductive health knowledge, behaviors, and outcomes [6]. Specific to this study, low health literacy has been linked to lack of contraception and fertility knowledge, including mechanisms of contraception methods [7] and contraception adherence [8].

The nature of self-reported datasets and secondary analysis leaves much room for information bias due to social desirability and health literacy levels. The potential for misunderstanding questions and response options impacts how respondents understand and reply to questions, which, in turn, may influence analyses that only utilize quantitative data and do not utilize additional qualitative data collected. The Pregnancy Risk Assessment Monitoring System (PRAMS) is a Centers for Disease Control and Prevention (CDC)-facilitated population-based surveillance system, which supports maternal and child health research [9]. An example of information bias is found in one PRAMS analysis, which found that among women reporting a mistimed or unwanted pregnancy as a result of contraception nonuse, 30% reported that they did not use a method because they did not mind having a pregnancy [10]. Without deeper investigation, it is not intuitive that someone would report both an unwanted pregnancy and not minding a pregnancy at the same time. Similarly, our original study sought to explore respondents' reported *postpartum* contraception behavior among those reporting an unintended pregnancy, including reasons why women do not actively choose subsequent pregnancy prevention. The objective of these analyses was to identify how responses to PRAMS survey questions match respondents' consistent and accurate (reliability and validity) interpretations through the additional analysis of qualitative observations [11,12].

## Methods

PRAMS, a surveillance project run by the CDC and state and local health departments, covers approximately 81% of US births by collecting state birth certificate data and including both universal

core questions and site-specific questions about maternal attitudes and behaviors related to pregnancy [9]. Postpartum women are identified through birth certificate data and mailed questionnaires 2 to 4 months after delivery. Potential participants are followed up via mail and phone to encourage higher response rates. PRAMS asks quantitative categorical questions, some of which include a response choice of "other," which is followed by an open text box to fill in another response. Thus, these open-text responses were used to qualitatively evaluate quantitative survey questions. For these analyses, the core questionnaire of interest asks participants about their reasons for not preventing a subsequent pregnancy through categorical and open-ended options [13]. We chose a subset of Phase VII of PRAMS (2012 to 2015) because it includes the most comprehensive list of categorical choices ($n = 9$) compared to Phases V, VI, and VIII. PRAMS Phase VII respondents ($n = 65,407$) were chosen via recent birth certificate data; those with high-risk characteristics were selected at a higher rate to collect adequate amount of data on smaller populations. Phase VII's response rate threshold was $\geq 60\%$. This study analyzes only the 2012 cycle as a standard sample that could both satisfy quantitative value and reach qualitative saturation.

We integrated qualitative and quantitative epistemologies to discover a pragmatic middle-ground reality rather than objective or subjective accounts (for example, fixed qualitative coding structures and interrater reliability) [14]. Balancing quantitative and qualitative rigor standards of sample size and theoretical saturation, we performed a secondary analysis of Phase VII PRAMS 2012 cycle ($n = 31,208$) to investigate contraception use/nonuse and reasons for nonuse. We first took a basic quantitative inventory of original response data percentages. We then used a qualitative thematic approach to develop our recategorization protocol. This protocol was used to standardize a subjective qualitative approach, creating a prospective plan to better fit the objective quantitative intentions of the PRAMS survey (Fig 1). Analysts documented the qualitative process with observational memos, including thoughts about sociolinguistics and phenomena occurring. Lastly, we used kappa scores, counts, and percentages to tie in qualitative epistemological findings with quantitative measures. This mixed-method lens enabled a richer understanding of the data while managing researcher biases. This research was exempt from review by Upstate Medical University's Institutional Review Board (#1165109).

## Original data analysis

To identify those that were and were not using contraception, we analyzed data from PRAMS core question 50: "Are you or your husband or partner doing anything now to keep from getting pregnant? Some things people do to keep from getting pregnant include using birth control pills, condoms, withdrawal, or natural family planning" ($n = 31,208$). We then explored possible postpartum contraception uptake barriers through core question 51: "What are your reasons or your partner's reasons for not doing anything to keep from getting pregnant postpartum? Check all that apply." In addition to structural and financial reasons for contraception nonuse, the term "barriers" includes personal constraints such as cultural and perception influence [15]. We calculated original postpartum contraception use prevalence ($n = 30,637$) and proportions of each categorical nonuse reason including abstinence, wants pregnancy, does not want to use, side effects, partner does not want to use, access problems, tubal ligation, vasectomy, currently pregnant, and other reason ($n = 6,044$) using SPSS software version 26 (IBM). Lastly, we reviewed case-by-case consistency between quantitative data and corresponding qualitative reports in Excel 2016 to improve categorization [16].

## Recoding variables

We followed traditional qualitative underpinnings of emerging and iterative data analysis to develop steps of a systematic approach to data recategorization [12]. Researchers read open-

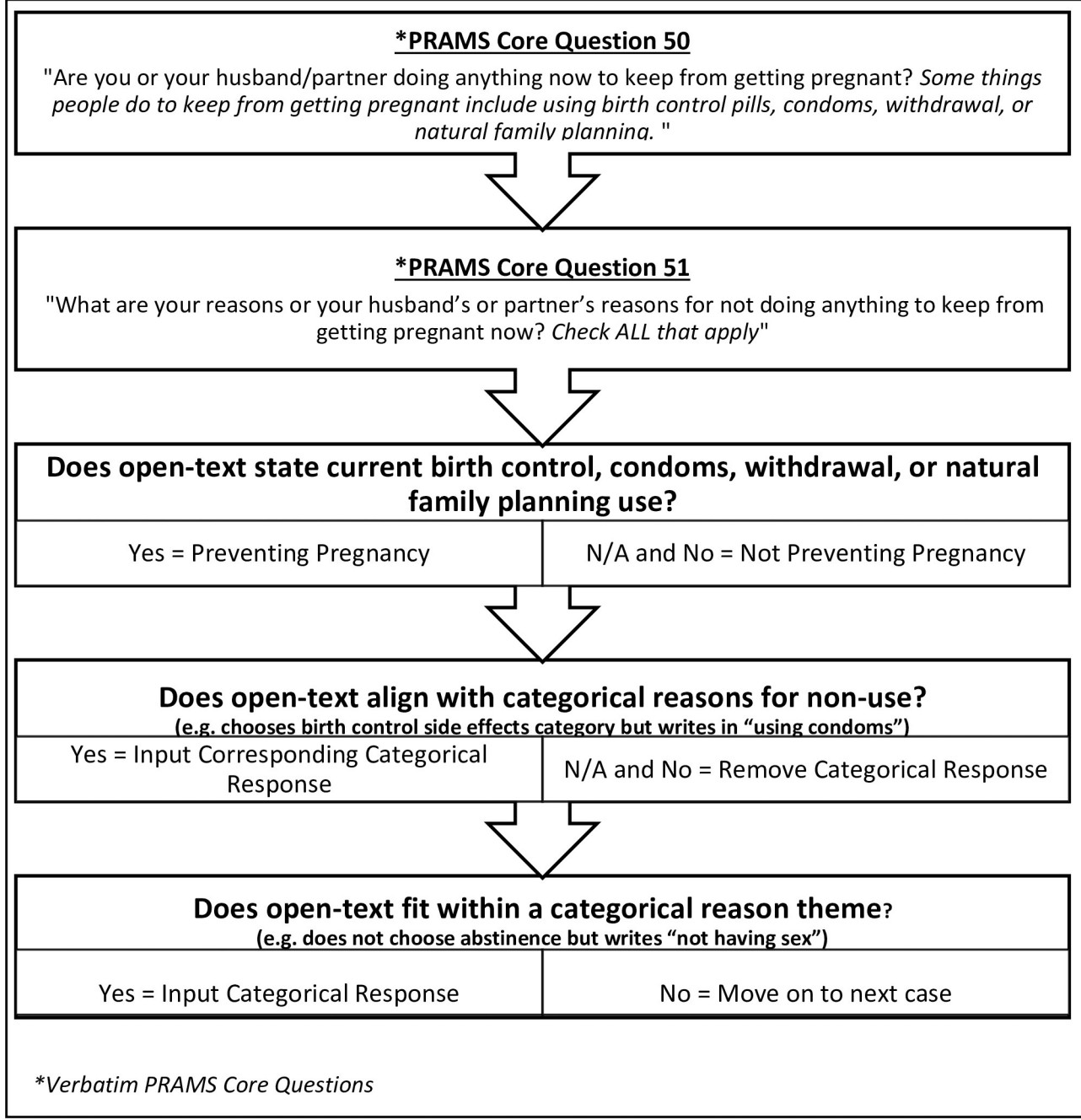

**Fig 1. Steps for qualitative recoding PRAMS Phase VII core question 50 and 51 via open-text responses.** Researchers used this tool to ensure consistent coding and recategorization of contraception behavior responses. NFP, natural family planning; PRAMS, Pregnancy Risk Assessment Monitoring System.

text responses for data familiarization, open-coded for preliminary trends, deliberated on a uniform coding protocol, and finalized the systematic recoding process (Fig 1). Data were manually recoded following predetermined steps. First, we reviewed the postpartum contraception use (y/n) variable according to the question framework, which did not categorize abstinence, vasectomy, and tubal ligation as contraception methods. We recategorized

respondents as using contraception if the open-text response indicated pregnancy prevention through examples given in PRAMS core question 50 (birth control, condoms, withdrawal, or natural family planning). Next, we deleted categorical nonuse reasons if they were contradicted by open-text responses and input categorical reasons if open-text responses aligned with predetermined categories. For example, if abstinence category was not chosen but a respondent wrote "not having sex," we checked the abstinence category. In addition, specific open-text responses that fit within larger categorical themes were appropriately coded in their corresponding category. For example, we categorized financial barriers or waiting for a contraception method as "having problems getting birth control," religious or cultural objections to birth control as "not wanting to use a method," and currently okay with becoming pregnant again as "I want to become pregnant." Using the agreed-upon codebook, original coding was conducted by one female author (NKR), creating a new variable for each code (Affordability; Beliefs; Breastfeeding: LAM*/No ovulation, Breastmilk Impact; Fertility Issues; Hysterectomy; Office Barriers; Same Sex Relationship; Situational Barriers; Medical Condition; Will Get a Method). We calculated number of respondents in each new category via SPSS descriptive statistics function. A separate female author (EB) recoded a random sample of 10% of all responses. A kappa score of agreement for interrater reliability was calculated for each variable. We calculated new unweighted and weighted prevalences to show a more accurate picture of postpartum contraception use and reasons for nonuse, using chi-squared tests to determine differences in variable distribution. Weighted percentages were calculated in Stata version 16.1 using survey weights and svy commands.

### Qualitative analysis

We explored open-text responses in NVivo version 12 to identify common and specific nonuse explanations not captured by categorical options. We reached theme development through content analysis, allowing us to become familiar with the data, identify initial codes, and subsume codes into categories [17]. We then quantified each theme to illustrate frequencies and exemplified these phenomena through excerpts. This study is reported as per the Standards for Reporting Qualitative Research (SRQR) guideline (S1 Checklist) [18].

## Results

### Original response data

Raw reports of postpartum contraception use showed that 78.8% ($n$ = 24,593/31,208) of postpartum women were using some form of postpartum contraception (Table 1). Further, many of those that reported no pregnancy prevention also reported controlling their fertility through abstinence (36.7%) or tubal ligation (13.2%) in the follow-up question. The most prominent reasons for not preventing pregnancy were personal preference to not use birth control (29.8%) and concerns with birth control side effects (27.9%). Respondents commonly chose more than one reason they were not using a pregnancy prevention method. Despite this question's 9 categorical choices, 30% reported additional reasons for not preventing a subsequent pregnancy via open-text responses.

Instead of choosing a categorical reason for not preventing a postpartum pregnancy, respondents sometimes alternatively reported an open-text response that fit within the existing response categories. Similarly, other open-text responses included explanations of categories already chosen (for example, choosing abstinence and elaborating on why they were abstinent).

Though the "reasons for contraception nonuse" question asked respondents that were using contraception to skip this section, respondents using a method sometimes still answered

**Table 1. Original and recoded postpartum contraception behavior (*n* = 31,208) and reasons for nonuse among PRAMS Phase VII (2012) respondents.**

|  | Original Response Data | | | Recoded Sample | | |
|---|---|---|---|---|---|---|
|  | n | % | Weighted % | n | % | Weighted % |
| Using Contraception Method |  |  |  |  |  |  |
| Yes | 24,593 | 78.8 | 79.0 | 24,020 | 77.0 | 77.2** |
| No | 6,044 | 19.4 | 19.3 | 6,719 | 21.5 | 21.4** |
| Missing | 571 | 1.8 | 1.8 | 469 | 1.5 | 1.4 |
| Nonuse Reasons* |  |  |  |  |  |  |
| Abstinence | 2,222 | 36.7 | 6.7 | 2,183 | 32.5 | 6.6** |
| Want Pregnancy | 990 | 16.4 | 3.2 | 1,029 | 15.3 | 3.4** |
| Do not Want To Use Birth Control | 1,801 | 29.8 | 6.0 | 1,818 | 27.1 | 6.1** |
| Birth Control Side Effects | 1,688 | 27.9 | 5.2 | 1,680 | 25.0 | 5.2** |
| Partner's Decision | 733 | 12.1 | 2.4 | 721 | 10.7 | 2.4** |
| Problems Getting Birth Control | 271 | 4.5 | 0.8 | 370 | 5.5 | 1.1** |
| Tubal Ligation | 797 | 13.2 | 2.4 | 794 | 11.8 | 2.4** |
| Vasectomy | 104 | 1.7 | 3.1 | 101 | 1.5 | 3.1** |
| Currently Pregnant | 187 | 3.1 | 4.6 | 186 | 2.8 | 4.6** |
| Other Barriers | 1,569 | 30.0 | 5.2 | 1,356 | 20.2 | 4.4** |

*Percent denominators are those not using contraception (original response data: 6,044; recoded sample: 6,719).

**p-Value < 0.001.

PRAMS, Pregnancy Risk Assessment Monitoring System.

this question. Some reported contraception use but provided reasons why they were not preventing pregnancy. In other cases, open-text responses described their current method (for example, IUD, implant, and birth control shot). Further, both the contraception user and nonuser groups reported practicing abstinence or having a tubal ligation procedure, suggesting a nonuniform understanding of pregnancy prevention methods.

## Recoded data

We found high interrater reliability between coders (kappa range 0.82 to 1.0) and slight changes in the prevalence of postpartum contraception use and nonuse reasons after our manual recoding (Table 1). Overall, 675 more women were not using a PRAMS-defined contraception method than the original data showed (21.5% versus 19.4%). Moreover, a 9.8% decrease (*n* = 213) in the "other reason for nonuse" category illustrates a sizable group that should have skipped this question (that is, those that used contraception or did not indicate contraception status). Likewise, most recoded categorical prevalences decreased from original data reports. All recoded variables were statistically significantly different when weighted percentages were compared to original percentages.

## Open-text qualitative themes

Open-text responses identified other nonuse themes in addition to prompted reasons for contraception nonuse (Table 2). We found that some women were preventing pregnancy through other contraceptive methods; 40 women did not require contraception because they were in a same-sex relationship, 32 had a hysterectomy after their last birth, and 197 had difficulty getting pregnant for several other reasons. Those with fertility problems often mentioned how long it took to conceive, conditions that impacted fertility, and fertility treatment methods. Others relied on not having a postpartum menstrual cycle (*n* = 100), with most of these

**Table 2. Qualitative themes of postpartum contraception nonuse reasons among PRAMS Phase VII (2012) respondents.**

| Theme | Examples | n |
|---|---|---|
| **Affordability** | *"No insurance," "My insurance doesn't believe in birth control!," "Too expensive"* | **67** |
| **Beliefs** | *"Religious reasons," "I don't believe in birth control," "All God's plan"* | **90** |
| **Breastfeeding** | | **164** |
| LAM*/No ovulation | *"Breastfeeding exclusively," "No menstruation due to breastfeeding," "Nursing is a form of birth control"* | 100 |
| Breastmilk Impact | *"Birth control almost stopped my supply of Breastmilk," "Chemicals/hormones breastfeeding"* | 31 |
| **Fertility Issues** | *"Took a really long time to get pregnant" "We used IVF"* | **197** |
| **Hysterectomy** | *"Womb removed," "Hysterectomy after birth," "Doctor took my uterus"* | **32** |
| **Office Barriers** | *"we were not informed," "my doctor doesn't prescribe birth control," "signed up for it but still waiting"* | **56** |
| **Same-Sex Relationship** | *"female partner!," "SAME-SEX PARTNER *maybe you should change your survey," "I'm gay"* | **40** |
| **Situational Barriers** | *"I can't find the right birth control," "Haven't had time," "Stress," "Transportation issues"* | **76** |
| **Medical Condition** | *"Blood pressure too high," "Blood clot disorder," "Can't take hormonal due to certain meds"* | **36** |
| **Will Get a Method** | *"I am getting tubes tied," "Waiting to get IUD put in," "husband is scheduling a vasectomy"* | **115** |

*Lactation amenorrhea method.

IVF, in vitro fertilization; IUD, intrauterine device; PRAMS, Pregnancy Risk Assessment Monitoring System.

women using lactation amenorrhea to prevent ovulation. Open-text responses showed frustration that these reasons were not listed as categorical options (for example, capitalizing words, using exclamation points, and suggesting survey question changes).

Women reported that they could not use a contraception method for a few common reasons. Approximately 36 women reported having medical conditions that limited their preferred birth control options (for example, heart and blood conditions). Similarly, breastfeeding mothers were concerned about birth control's breastmilk impact ($n = 31$), such as hormones and decreased milk supply. Others chose not to use a method due to negative contraception beliefs ($n = 90$), which often stemmed from cultural and religious values.

Many participants intended to use a contraception method in the near future but were not currently preventing pregnancy for a number of reasons ($n = 115$). Affordability was a concern for 67 women, with several reporting no contraception coverage because they lost their insurance during the postpartum period or their insurance did not cover contraception. Further, 56 women reported office barriers, such as long waits for contraception counseling and procedure appointments, and clinician reluctance to provide a method. Further, 76 women faced situational barriers to contraception, including lack of transportation or time to attend an appointment.

## Discussion

Our study shows that qualitative analyses of open-text survey responses can contribute to accurate data categorization and resulting analyses. Indeed, while several overall weighted percentages in each category were the same or only different by tenths of a percent, chi-squared tests indicate that the recoded categories were all statistically significantly different after this recoding exercise. The statistically significant differences suggest the importance of

reexamining survey responses to improve data quality; however, it should be noted that the original and recoded findings yield similar measures of contraception use. Overall, these PRAMS questions illuminated an unclear understanding of the definition of pregnancy prevention in some instances, and this may have contributed to inconsistent answers based on respondent interpretation. Additionally, the nature of self-administered mailed surveys may have contributed to respondents missing skip patterns (that is, some that *were* preventing pregnancy mistakenly reported reasons why they were *not* preventing pregnancy). With many respondents reporting similar noncategorical reasons for contraception nonuse (for example, breastfeeding and fertility issues), additional categorical options could be useful to capture a more comprehensive picture of contraception behavior.

Using PRAMS categorical and open-text responses to identify contraception behaviors and barriers enabled the study team to collate population-based phenomena occurring among postpartum women. This study is unique in its ability to contribute qualitative information about a large sample of postpartum women's contraception behavior. Further, the methods used establish qualitative data's usefulness in large datasets. Qualitative thematic text analyses are already used to explore *new* ideas within open-ended survey questions [19]. That is, researchers may conduct qualitative methods on either (1) a stand-alone open-ended survey question or (2) a categorical survey question that is followed by a prompt to "explain" the categorical choice. However, this paper offers a detailed process of how qualitative and quantitative methods can be used in combination as a tool to strengthen already existing survey questions.

Our unique qualitative approach to this dataset identified misclassification bias, including miscategorized categorical responses and underlying contraception barriers not captured by original data analysis. We mitigated manual coding error and researcher biases by limiting coding to 1-hour periods and calculating kappa scores for codebook validation. The qualitative portion and systematic review of original data uncovered detailed information about respondent values and behavior rather than relying solely on self-reported quantitative data. Though we mitigated biases when possible, qualitative methods innately introduce researcher bias. Further, the use of these methods for survey improvement were only performed on one survey question within one national dataset, leaving a need for replication in other survey open-text datasets to better validate as a transferrable approach.

Based on our PRAMS analyses, we suggest collecting cleaner data, ensuring that questions like these are asked in a way that systematically categorizes contraception and uses simple language comprehendible to those with limited health literacy or language barriers. For example, the National Survey of Family Growth collects information about using contraception methods by asking yes–no questions about every method [20]. While we understand each survey's scope and data collection method can contribute to the comprehensiveness of contraception use questions, we recommend open-text responses like this PRAMS question are thoroughly examined and cleaned prior to secondary analyses.

Specific to PRAMS question 50 and 51, we recommend asking all respondents "Are you doing any of the following to prevent pregnancy?", providing a comprehensive list of all contraception methods with yes/no responses, and including a "none of the above" option. If a respondent chose none of the above, they would then answer the second question "What are your reasons for not preventing pregnancy?" Fig 2 displays an example of a comprehensive survey design approach to eliciting more accurate contraception behavior approaches.

PRAMS and other family planning research initiatives may consider our qualitative themes for possible contraception nonuse barriers. Not only did qualitative inquiry identify additional barriers, but we found respondent frustration with the exclusion of same-sex partners, religious and cultural values, and those struggling with infertility. Our qualitative findings can be used to explore phenomena such as access barriers and the breastfeeding–contraception

1. Are you or your partner doing any of the following to keep from getting pregnant?

|  | Yes | No |
|---|---|---|
| **Abstinence** |  |  |
| **Same-sex or non-penis-in-vagina sex** |  |  |
| **Tubal Ligation** |  |  |
| **Vasectomy** |  |  |
| **Exclusive Breastfeeding** |  |  |
| **Natural (Track Cycle/Rhythm Method)** |  |  |
| **Withdrawal (Pull Out Method)** |  |  |
| **Spermicide** |  |  |
| **Condoms** |  |  |
| **Birth Control Pills** |  |  |
| **Birth Control Patch** |  |  |
| **Birth Control Ring** |  |  |
| **Birth Control Shot** |  |  |
| **Diaphragm or Cervical Cap** |  |  |
| **Implant (Rod in arm)** |  |  |
| **IUD** |  |  |
| **None of Above** |  |  |

**[If survey participant choses "none of above"]**

2. What are your reasons or your partner's reasons for not doing anything to keep from getting pregnant?

|  | Yes | No |
|---|---|---|
| **I don't mind if I get pregnant** |  |  |
| **I don't want to use birth control** |  |  |
| **My partner doesn't want to use anything** |  |  |
| **I have had a hysterectomy** |  |  |
| **I have side effects from birth control** |  |  |
| **I'm having problems getting birth control from my provider** |  |  |
| **I cannot afford birth control** |  |  |
| **I don't believe in using birth control** |  |  |
| **We have problems getting pregnant** |  |  |
| **Birth control interferes with a medical condition** |  |  |
| **Other Barriers →Please Tell Us:** |  |  |

**Fig 2. Example of comprehensive contraception behavior survey questions.** Revising current contraception behavior survey questions to a checklist approach may better engage the participant to elicit more accurate and detailed information. IUD, intrauterine device.

relationship. We encourage these categories to be more inclusive of gender, sexual orientation, and culture. Further, misreports found in the dichotomized pregnancy prevention question can be eliminated by asking all individuals about each specific contraception behavior and provide a richer understanding of pregnancy prevention.

In conclusion, rigorous standards of both qualitative and quantitative epistemologies can be used as a mixed-method approach to strengthen survey methodology. The depth and breadth of this developed protocol may lead to more accurate findings and resulting health interventions that are more specific to the circumstances surrounding themes identified within participants' unprompted responses.

## Supporting information

**S1 Checklist. Standards for Reporting Qualitative Research (SRQR) [18].**
(PDF)

## Acknowledgments

We would like to thank the PRAMS Working Group for data collection and review of this article.

## Author Contributions

**Conceptualization:** Nicole K. Richards, Christopher P. Morley, Martha A. Wojtowycz, Brooke A. Levandowski.

**Data curation:** Nicole K. Richards, Martha A. Wojtowycz, Brooke A. Levandowski.

**Formal analysis:** Nicole K. Richards, Erin Bevec.

**Investigation:** Nicole K. Richards, Brooke A. Levandowski.

**Methodology:** Nicole K. Richards, Christopher P. Morley, Martha A. Wojtowycz, Brooke A. Levandowski.

**Project administration:** Brooke A. Levandowski.

**Resources:** Martha A. Wojtowycz, Brooke A. Levandowski.

**Supervision:** Christopher P. Morley, Martha A. Wojtowycz, Brooke A. Levandowski.

**Validation:** Christopher P. Morley, Brooke A. Levandowski.

**Visualization:** Nicole K. Richards, Erin Bevec.

**Writing – original draft:** Nicole K. Richards, Brooke A. Levandowski.

**Writing – review & editing:** Nicole K. Richards, Christopher P. Morley, Martha A. Wojtowycz, Erin Bevec.

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
