## [Editor Report · Decision Letter 0]

22 Mar 2021

Dear Dr Richards, 

Thank you for submitting your manuscript entitled "Using qualitative inquiry to improve survey data collection: An evaluation of PRAMS postpartum contraception behavior questions" for consideration by PLOS Medicine.

Your manuscript has now been evaluated by the PLOS Medicine editorial staff and I am writing to let you know that we would like to send your submission out for external peer review.

Kind regards,

Caitlin Moyer, Ph.D.

Associate Editor

PLOS Medicine

---

## [Decision Letter · Decision Letter 1]

23 Jun 2021

Dear Dr. Richards,

Thank you very much for submitting your manuscript "Using qualitative inquiry to improve survey data collection: An evaluation of PRAMS postpartum contraception behavior questions" (PMEDICINE-D-21-00947R1) for consideration at PLOS Medicine. 

Your paper was evaluated by a senior editor and discussed among all the editors here. It was also discussed with an academic editor with relevant expertise, and sent to three independent reviewers. The reviews are appended at the bottom of this email and any accompanying reviewer attachments can be seen via the link below:

[LINK]

In light of these reviews, I am afraid that we will not be able to accept the manuscript for publication in the journal in its current form, but we would like to consider a revised version that addresses the reviewers' and editors' comments. Obviously we cannot make any decision about publication until we have seen the revised manuscript and your response, and we plan to seek re-review by one or more of the reviewers. 

We expect to receive your revised manuscript by Jul 14 2021 11:59PM. Please email us (plosmedicine@plos.org) if you have any questions or concerns.

We look forward to receiving your revised manuscript. 

Sincerely,

Caitlin Moyer, Ph.D.

Associate Editor 

PLOS Medicine

plosmedicine.org

1. Abstract: Please combine the Methods and Findings sections into one section, “Methods and findings”.

2. Abstract: Methods and Findings: Please provide some additional details on the PRAMS survey, including the individuals represented and setting (e.g. the US).

3. Abstract: Methods and Findings: In the last sentence of the Abstract Methods and Findings section, please describe the main limitation(s) of the study's methodology.

4. Abstract: Conclusions: Please address the study implications both for the PRAMS questionnaire and if possible the implications for other similar surveys that may be used to evaluate postpartum pregnancy prevention use or non-use.

5. Author summary: At this stage, we ask that you include a short, non-technical Author Summary of your research to make findings accessible to a wide audience that includes both scientists and non-scientists. The Author Summary should immediately follow the Abstract in your revised manuscript. This text is subject to editorial change and should be distinct from the scientific abstract. Please see our author guidelines for more information: https://journals.plos.org/plosmedicine/s/revising-your-manuscript#loc-author-summary

6. As noted by Reviewer 1, the survey data seem to be relatively out of date. Please update with more recent data.

7. Methods: Thank you for including the SRQR checklist. Please add the following statement, or similar, to the Methods: "This study is reported as per the Standards for Reporting Qualitative Research (SRQR) guideline (S1 Checklist)."

8. Methods: Did your study have a prospective protocol or analysis plan? Please state this (either way) early in the Methods section.

9. Methods: Please include the relevant PRAMS questions as a supporting information file, if possible (or please indicate if these are Core Question 50 and 51 that are provided in Figure 1).

10. Findings: Please rename this section “Results”

11. Results: It may be helpful to provide a table with relevant demographic/characteristics of the study population.

12. Results: Line 118-119: Rather than referring to “numerous respondents” it would be helpful here to provide the number/percentage. Similarly, this could be changed at line 123 where “many respondents” are indicated.

13. Discussion: Please present and organize the Discussion as follows: a short, clear summary of the article's findings; what the study adds to existing research and where and why the results may differ from previous research; strengths and limitations of the study; implications and next steps for research, clinical practice, and/or public policy; one-paragraph conclusion.

14. Line 206-207: Please add the ethical approval information to the Methods section.

15. References: Please use the "Vancouver" style for reference formatting, and see our website for other reference guidelines https://journals.plos.org/plosmedicine/s/submission-guidelines#loc-references

16. Table 1: Please make it clear in the second part of the table, that the denominators for the Non-Use reasons are 6044 for the “Raw Sample” and 6719 for the “Recoded Sample” if accurate. Please consider changing “raw” to “original” or similar, to clarify the meaning.

17. Figure 1: Please spell out the abbreviation NFP in the figure or legend.

18. SRQR Checklist: Please use section and paragraph numbers, instead of page and line numbers, to refer to locations within the manuscript.

Comments from the reviewers:

Reviewer #1: 

Although the topic is important and I applaud the authors for examining qualitative data to reduce measurement error and make recommendations to strengthen surveillance in the future, I had some concerns as well as recommendations to improve the paper. 

General

 * When linking pospartum contraception with outcomes (e.g., UIP), make sure you add "use" - since postpartum contraception must be used to impact outcomes.

 * PRAMS data are population-based but not nationally representative (although they represent a large proportion of US births). Please correct language in paper. 

 * The paper would benefit from editing.

 * The data are quite outdated - from 2012. 2019 PRAMS data are now available. The findings would be more helpful if the analysis were re-done using the most updated data available.

 * Contraceptive use behaviors are complex and have to be analyzed carefully. This includes data from PRAMS as well as other data systems (e.g., BRFSS). It is quite common for someone to report not using a method, but then listing a method as a reason for not using contraception (e.g. tubal ligation or partner sterilization). This isn't an issue with question reliability; respondents just don't think that something they don't do daily constitutes contraception use. So recoding is always recommended.

 * As a PRAMS analyst, I am a proponent of examining the write-in responses and coding as appropriate. But for a publication in the peer-reviewed literature, I am more interested in how such efforts change the prevalence in the behavior - (especially one that is tracked for Healthy People). As such, I wish the authors would apply the recodes of reasons for non-contraception use and apply then to the contraceptive use variable (e.g, those using tubal ligation, vasectomy would be considered contraceptive users). 

Abstract

 * Change 2nd sentence to reflect "…..postpartum contraceptive use and reasons for non-use" (rather than postpartum contraception non-use and reasons for not preventing pregnancy. Not sure what is meant by "not preventing pregnancy". 

 * Consider making the 3rd sentence specific to PRAMS. Something like "In addition to pre-determined, closed-ended response options on reasons for not using contraception, respondents could write in a response. This qualitative data can be used….

 * I'm not sure about "analysis of the write-in responses validates PRAMS". It does reduce misclassification bias since many response options can be recoded to reflect contraceptive use (e.g., when a woman reports partner vasectomy). So it improves the performance/measurement of the indicator, but to validate, you would need to compare against something else (like electronic medical records). 

 * PRAMS is a complex survey and weights must be applied to interpret estimates. It would be nice to see how the weighted proportions of non-use differed comparing raw versus recodes responses.

 * Typically, one would recode responses of "tubal ligation" as current contraceptive use (rather than non-use, at low risk for pregnancy). When you mention "abstinence" do you mean period abstinence/calendar method (which is a contraceptive method); or not being sexually active? 

Introduction

 * Page 2, line 52 - I understand the concern about self-report leading to bias, but how does the size of the dataset lead to bias?

 * Not following page 2, lines 53-56 - how is this an example of bias? The cited report examined a question on contraception use at time of conception. Please make this clearer, since it differs from the main question of interest for this paper - postpartum contraceptive use. 

 * Page 2, line 57 - "why women do not prevent a subsequent pregnancy". The PRAMS question examine postpartum contraceptive use. Keep the focus on the behavior at hand. We do not know if they go on to become pregnant again or not. 

Methods

 * I still think there is value in updating with more recent data. If you wish to look at Phase 7 because it includes more response options, at least examine the most recent Phase 7 data which is 2015. 

 * Didn't quite follow page 3, lines 70-73.

 * Page 3, line 77 - not all reasons for 'not using contraception' are barriers - they are often personal choice. Consider rephrasing to something that more closely matches what was measured (e.g., reasons for not using contraception at time of survey). 

Results

 * Table 1 - to me, the differences between raw and recoded estimates are not that striking for postpartum contraception use. For reasons for non-use, the proportion reporting 'other barriers' is much lower after recoding, which makes sense; but I would expect the proportions in the recoded sample to be higher compared with the raw sample (due to recoding of write-in other responses), but for most, it is lower. Why is this? Since PRAMS data should be weighted, how did the proportions compare (examining raw vs recoded) once the proportions were weighted. 

 * Table 2 findings are interesting and illuminating. To put into perspective however, these responses represent a very small proportion of PRAMS respondents (<3%). 

 * Figure 1 - 4th row of boxes - change "preventing pregnancy" and "not preventing pregnancy" to "using contraception" and "not using contraception". Contraception is not 100% effective, so using contraception does not translate to preventing pregnancy (since pregnancies can still occur).

 * Figure 1 - can you explain more what is meant by "does open text align with categorical reason for non-use"? Maybe provide some examples of when text does not align? I would also recommend a last step in Figure 1 - apply reasons for non-contraceptive use that are contraceptive methods (tubal ligation/partner vasectomy) to recode the contraceptive use (yes/no) variable. 

 * Figure 2, Table 1 - for "abstinence" - it's important to distinguish between 'not having sex' a

 * Page 5, lines 126-127 - because these are postpartum women, women may not yet be sexually active. But because sexual activity will likely resume in the near future, it is common practice for women to start a contraceptive method before resuming sexual activity. So I'm not sure how much weight I would give to those types of discrepancies. A good example is a woman who starts an IUD or implant in the hospital before discharge but reports being abstinent early postpartum because she has not yet resumed sexual activity. 

Discussion

 * Lines 161-162 - although discrepancies were found, it was only a very small proportion (<3%). This sentence seems a bit strong. Same comment to page 8, line 191. Can you really say "many" - from my count it was 873/31,208 or 2.8%. Perhaps I am calculating incorrectly?

 * Line 164 - change "preventing pregnancy" to "using contraception" - this should be made throughout. Again, per my comment above, this is not that surprising and even expected given the population is postpartum. If examining BRFSS data, then you'd have a stronger case. 

 * Line 168-169 - although I appreciate the shout out to NSFG which is an amazing data source, I'm not sure how helpful these concluding remarks/conclusions are. NSFG is a survey largely focused on fertility (so more attention can be given to contraceptive use). It is also administered in person so quality of data is higher. PRAMS is a general survey on maternal behaviors and experiences, and postpartum contraceptive use is only a very small component. It's mode of data collection is different than NSFG.

Reviewer #2: This is a well written descriptive study on an important topic. The manuscript will no doubt help researchers in accurately measuring contraceptive prevalence, taking into account the reasons stated for contraceptive non-use. 

I have a few points to note. First is; should vasectomy, tubal ligation, abstinence not be considered as contraceptive methods? If the answer is yes, I suppose that contraceptive prevalence is underestimated in PRAMS. That takes me to my second point, I have always believed that contraceptive prevalence is underestimated in large surveys. This manuscript further supports my observation. 

I think authors should reflect on how the lack of proper recoding and cleaning of data may result in underestimation of contraceptive prevalence in their discussion. 

Reviewer #3: This brief manuscript investigates the effects of interpreting and back-coding open-ended survey responses on population estimates from the PRAMS Pregnancy Risk Monitoring System. For each variable code examined, this exercise altered response distributions, although, thankfully, not to an appreciable extent. Nonetheless, the analyses presented reflect improved estimates and might suggest to the PRAMS data management team that they consider routine application of these or similar back-coding practices when finalizing public use versions of PRAMS data. 

The analyses also generated recommendations for revision to the block of PRAMS questions that ask about reasons for not engaging in postpartum contraception. The most important of these recommendations is that the main item ("Are you or your partner doing any of the following to keep from getting pregnant?") be revised from a Check-all-that-apply format to one that asks respondents to answer "yes" or "no" for each response option. Structuring these questions as recommended by these authors is in fact a piece of convention wisdom in the survey methodology literature and it is somewhat surprising that PRAMS is not already doing so. Hopefully, this research will encourage them to consider doing so. One suggestion, however, in the author's recommended reformatting, which is presented in Figure 2, is to remove the "Check all that apply" instruction from the question stem, as that ironically harkens back to the previous format, which they are recommending be revised. Indeed, the "check all that apply" instruction may encourage some respondents not to follow the yes-no response format, so I strongly recommend it be removed here. Otherwise, this is a well written and useful research note, one that employs statistical methods appropriate for the data and research questions. I believe it has the potential to make a good contribution to the PRAMS research process.

[LINK]

---

## [Decision Letter · Decision Letter 2]

26 Aug 2021

Dear Dr. Richards,

Thank you very much for submitting your manuscript "Using qualitative inquiry to improve survey data collection: An evaluation of PRAMS postpartum contraception behavior questions" (PMEDICINE-D-21-00947R2) for consideration at PLOS Medicine. 

Your revised paper was evaluated by a senior editor and discussed among all the editors here. It was also discussed with an academic editor with relevant expertise, and sent back to one of the reviewers. The reviews are appended at the bottom of this email and any accompanying reviewer attachments can be seen via the link below:

[LINK]

In light of the reviewer feedback, I am afraid that we will not be able to accept the manuscript for publication in the journal in its current form, but we would like to consider a revised version that addresses the reviewers' and editors' comments. Obviously we cannot make any decision about publication until we have seen the revised manuscript and your response, and we plan to seek re-review by one or more of the reviewers. 

We expect to receive your revised manuscript by Sep 16 2021 11:59PM. Please email us (plosmedicine@plos.org) if you have any questions or concerns.

We look forward to receiving your revised manuscript. 

Sincerely,

Caitlin Moyer, Ph.D.

Associate Editor 

PLOS Medicine

plosmedicine.org

1. Please fully address the remaining points mentioned by Reviewer 1.

2. Title: Please revise your title according to PLOS Medicine's style. Your title must be nondeclarative and not a question. It should begin with main concept if possible. "Effect of" should be used only if causality can be inferred, i.e., for an RCT. Please place the study design ("A randomized controlled trial," "A retrospective study," "A modelling study," etc.) in the subtitle (ie, after a colon).

3. Abstract: Line 20-21: We suggest removing “indicating” and replacing with “potentially indicating” or “suggesting” as this is an interpretation.

4. Abstract: conclusions: Line 24-25: We suggest noting “Respondents reporting contraception non-use often described in open-text responses other methods of pregnancy prevention…”

At Line 27-28, we suggest clarifying this sentence: “Our findings contribute to the understanding of how open-text survey questions can lead to more accurate depiction of postpartum contraceptive behavior.” or similar.

5. The Author Summary should be formatted into 3 sections (Why was this study done; What did the researchers do and find; What do these findings mean), with 2-3 bulleted points per section. Please see our author guidelines for more information: https://journals.plos.org/plosmedicine/s/revising-your-manuscript#loc-author-summary

6. Introduction: The objectives presented in the final paragraph could be presented more clearly. It would be helpful if the introduction went into more detail on the significance of the study objective (such as potential for misinterpretation of questions/response options, reasons falling beyond categorical options presented; and how open-text response data may be informative in combination with quantitative data).

7. Methods: Line 76. Please report the number of individuals included in the PRAMS survey/included in this study. Please fully describe the population and setting. Please note whether all postpartum women were mailed the questionnaires, and provide details on how a sample was selected from birth certificate data for inclusion in the PRAMS. Please note survey response rates. Please note early on whether those included are nationally representative of the population of post-partum women in the United States.

8. Methods: Line 82: Please note the year(s) of the Phase 7 PRAMS.

9. Methods: Line 84-89: Please provide more description of the background for this process, keeping in mind this would be intended for a broad clinical/medical audience.

10. Methods: Original data analysis: Lines 91-103: Please mention here where the open-text responses are incorporated/evaluated, including the numbers of qualitative responses.

11. Methods: Line 112: Please note if the “PRAMS examples” indicates the categorical response options provided.

12. Methods: Please note in the Methods if your study had a prospectively developed analysis plan.

Did your study have a prospective protocol or analysis plan? Please state this (either way) early in the Methods section.

13. Methods: Qualitative analysis: Please completely describe the conceptual framework underlying your qualitative analysis.It would be helpful to provide more description on this process, keeping in mind a broad, general audience.

14. Results: Line 133: It is slightly confusing to refer to this as “original sample” and could be renamed “original response data” or similar.

15. Results: Please mention the total numbers included in the sample. Please present numerators and denominators for percentages.

16. Results: Line 140: Please clarify if the 30% reporting additional reasons indicates open-text responses.

17. Results: Line 153: Please provide the number of responses that were recoded based on the open-text responses.

18. Discussion: The organization of the discussion could be further refined. Please present and organize the Discussion as follows: a short, clear summary of the article's findings; what the study adds to existing research and where and why the results may differ from previous research; strengths and limitations of the study; implications and next steps for research, clinical practice, and/or public policy; one-paragraph conclusion.

Please begin with a statement reflecting the main findings of the study, rather than beginning with interpretations (“...illuminated an unclear understanding…”). We suggest expanding on the discussion of existing literature, and limitations of the study.

19. Line 186: Please clarify the sentence describing “respondents missing skip patterns”

20. Line 236: Please move this statement to the Methods, including the exemption number.

21. Line 238: Please remove the Consent for Publication statement.

22. Line 239: Please remove this section, and ensure all information on data availability are included in the manuscript submission metadata.

23. Lines 242-249: Please remove the Competing Interests, Funding, and Author contributions sections from the main text, and check that this information is complete and accurate in the relevant sections of the manuscript submission metadata.

24. Table 2: Please include the definition of all abbreviations in the legend (LAM).

25. Figure 2: Please define the abbreviation SP.

Comments from the reviewers:

Reviewer #1: * PRAMS data are population-based but not nationally representative (although they represent a large proportion of US births). The text (e.g., abstract) still refers to PRAMS as being nationally representative which is incorrect. 

 * PRAMS Phase 7 was conducted 2012-2015; PRAMS Phase 8 began in 2016 and 2019 data are now publicly available with 2020 data expected in the coming months. I can understand why the authors do not wish to replicate the work using Phase 8 2019 data given this would be very time intensive, but without doing so, I'm not sure how informative the report is related to the last sentence of the abstract: "This PRAMS question can be revised to improve clarity, including a more comprehensive list of pregnancy prevention methods and non-use options." Perhaps drop that sentence and make the conclusions a bit more broad - something along the lines of how examining write-in response can improve survey quality to inform both PRAMS and other similar surveys. 

 * I do not agree with the author's response to not present weighted estimates of contraceptive non-use. Estimates of PRAMS data should always be weighted so the estimates are population-based. Of course the in-depth examination of write-in responses is done at the individual level, but for one to meaningfully compare how different or similar estimates of non-use are comparing original/raw vs recoded data, those estimates should be weighted. Otherwise the reader cannot tell if the recoding had a meaningful impact on proportions. 

New sentence added to the discussion in response to Reviewer #2 seems a bit off to the reviewer's intent. Rather than "A failure to capture contraception behaviors and methods like abstinence and sterilization may contribute to an underestimate of those preventing pregnancy"….I suggest framing it around the importance of data cleaning/recoding and examining write-in response to improve data quality/validity. Because in reality, PRAMS will not be able to add new response options. To add a new response option, another one must come off given space limitations on the survey (PRAMS is hard copy vs BRFSS which is phone-administered and not every response is read by the interviewer). PRAMS questions deemed to have too many possible response options to be valid might instead be removed from the survey altogether, which I believe would be an unintended consequence of the paper. So careful framing is needed. The data aren't junk - analysts should just carefully examine write-in responses and recode appropriately.

[LINK]

---

## [Decision Letter · Decision Letter 3]

12 Nov 2021

Dear Dr. Richards,

Thank you very much for re-submitting your manuscript "Improving pregnancy prevention behavior data collection: a mixed-method inquiry of PRAMS postpartum contraception questions" (PMEDICINE-D-21-00947R3) for review by PLOS Medicine.

I have discussed the paper with my colleagues and the academic editor and it was also seen again by one of the reviewers. I am pleased to say that provided the remaining editorial and production issues are dealt with we are planning to accept the paper for publication in the journal.

[LINK]

We look forward to receiving the revised manuscript by Nov 19 2021 11:59PM.   

Sincerely,

Caitlin Moyer, Ph.D.

Associate Editor 

PLOS Medicine

plosmedicine.org

Requests from Editors:

1. Please address the points of Reviewer 1, including avoiding referring to the PRAMS survey as national (e.g at line 4, 67, and throughout) and tempering conclusions to reflect that original and re-coded findings lead to similar estimates from a clinical perspective, but noting that examination of responses could be useful for improving data quality.

2. Title: We suggest revising to "Use of open-text responses to recode categorical survey data on postpartum contraception use among women in the United States: A mixed-methods inquiry of Pregnancy Risk Assessment Monitoring System data" or similar, but please include "United States" in the title.

3. Abstract: Line 4: Please change this to “Pregnancy Risk Assessment Monitoring System (PRAMS).

4. Abstract: Line 24-25: We suggest revising the sentence to: “In this study, we observed that respondents reporting contraception non-use…” or similar.

5. Author Summary: Please organize the author summary into sets of 2-3 single sentence bullet points for each of the following questions. For an example, please see: https://doi.org/10.1371/journal.pmed.1002416

Descriptions of each of the three questions are included below. For example, the first two points could fall under the first question, the third, fourth, and fifth points could go under the second question, and the sixth and seventh points could go under the third question.

-Why Was This Study Done? Authors should reflect on what was known about the topic before the research was published and why the research was needed.

-What Did the Researchers Do and Find? Authors should briefly describe the study design that was used and the study’s major findings. Do include the headline numbers from the study, such as the sample size and key findings.

-What Do These Findings Mean? Authors should reflect on the new knowledge generated by the research and the implications for practice, research, policy, or public health. Authors should also consider how the interpretation of the study’s findings may be affected by the study limitations.

6. Author summary: Please revise the following bullet points:

-”We used qualitative methods to uncover richer information about respondents’ postpartum contraception behavior and interpretations of the survey question” -please remove “richer” or replace with “more detailed information” or similar.

-We suggest revising the next point to: “We observed that qualitative responses added to and altered quantitative contraception behavior reports, and that reasons for contraception non-use are incompletely captured by existing response categories.” or similar.

-We suggest revising this point to: “Family planning professionals may be able to obtain significant and detailed information about contraceptive behavior from open-ended responses, and this information may be useful to adjust clinical approaches to postpartum contraception conversations.” or similar.

7. Introduction: Line 67-68: Please refer to PRAMS as a population-based rather than national surveillance system.

8. Introduction: Line 75-77: Please revise to: “The objective of these analyses was to identify how responses to PRAMS survey questions match respondents’ consistent and accurate (reliability and validity) interpretations through the additional analysis of qualitative observations.” or similar, to clarify.

9. Methods: Line 79: Please revise the following to provide more relevant information on the PRAMS project (https://www.cdc.gov/prams/index.htm):

“PRAMS, a surveillance project run by the Centers for Disease Control and Prevention (CDC) and state and local health departments, collects state birth certificate data and includes both universal core questions and site-specific questions about maternal attitudes and behaviors related to pregnancy [reference].” or similar.

Please mention the fact that PRAMS covers approximately 81% of births in the United States (or please cite the most accurate estimate from the CDC).

10. Methods: Line 80: Please provide more details on how postpartum women were selected to be mailed the questionnaires (e.g. it seems as if questionnaires are sent 2-4 months after delivery, to a sample drawn monthly from current birth certificate data).

11. Methods: Line 86: In the abstract, PRAMS 2012 data are mentioned, and here a range of 2012-2015 is mentioned. Please clarify the year during which survey data were collected for this analysis.

12. Methods: Line 110: Please check if this should be “...postpartum contraception uptake barriers…”

13. Methods: Line 117: We suggest noting that “Original postpartum contraception use prevalence data were analyzed using SPSS software version 26 (IBM).” or similar. Please similarly clarify this for the mention of Excel (Line 119), Stata (Line 145) and NVivo (Line 148).

14. Results: Line 181: Please remove “...suggesting the importance of recoding.” as this is an interpretation.

15. Discussion: As Reviewer 1 points out, please temper conclusions to mention and reflect on the fact that the original and recoded findings are significantly different statistically, but are similar, in terms of clinical implications but that the examination of responses could be useful for improving data quality.

16. Discussion: Line 215: We suggest revising to: “...this suggests that additional categorical options could be useful to capture a more comprehensive picture of contraception behavior.”

17. Discussion: Line 218: We suggest removing the word “national” and replacing with “population-based” or similar.

18. Discussion: Line 234: Please avoid the word “national” and replace with “population-based” or similar.

19. Discussion: Line 260: Please revise to “may lead to” in the sentence.

20. References: Please provide the complete citation information for ref 14.

21. Table 1: Please do not report "p<0.05" and instead provide exact p values for each analysis in the weighted % column (including p values for results not reaching statistical significance).

22. Figure 1: There seem to be two versions of this figure. Please include a descriptive legend in addition to the title, for this figure. Please indicate natural family planning (NFP) in the legend.

23. Figure 2: Please include a descriptive legend in addition to the title for this figure. Please define all abbreviations used in the legend (IUD).

24. SRQR Checklist: Please refer to Methods, paragraph 2 as the location for IRB/ethical review information. Please also refer to Competing Interests and Financial Disclosures sections as locations for Conflict of Interest and Funding information.

Comments from Reviewers:

Reviewer #1: PRAMS data are population-based but not national (although they represent a large proportion of US births); data are state/site-specific not national. Abstract line 4 still refers to PRAMS as being national. 

I appreciate the authors comparing weighted proportions of original vs recoded variables; however, although statistically different via chi-square tests (page 8, lines 208-209), the differences are not meaningful from a clinical or public health surveillance perspective. When I look at Table 1 - I feel reassured that both methods (original vs recoded) produce very similar estimates of postpartum contraception use (79% vs 77%). That's an OK conclusion to have as well; you could still encourage analysts to examine write-ins (time permitting) to improve data quality.

[LINK]

---

## [Editor Report · Decision Letter 4]

29 Nov 2021

Dear Dr Richards, 

On behalf of my colleagues and the Academic Editor, Sarah Stock, I am pleased to inform you that we have agreed to publish your manuscript "Use of open-text responses to recode categorical survey data on postpartum contraception use among women in the United States: A mixed-methods inquiry of Pregnancy Risk Assessment Monitoring System data" (PMEDICINE-D-21-00947R4) in PLOS Medicine.

We also request that you address the following editorial issues:

1. Author summary: Please revise and re-organize the points of the Author Summary into three sections as follows: 

Why was this study done?

-Survey respondents can occasionally provide open-ended answers that may be overlooked during quantitative data analysis

- This study was conducted to study how qualitative inquiry of open-text responses to a survey question related to postpartum contraception use can be used to provide insight into data accuracy within quantitative surveys.

What did the researchers do and find?

-Using a mixed-method approach, we calculated descriptive statistics from PRAMS survey responses from 31,208 postpartum women in the US who provided categorized reasons for not using contraception and qualitatively analyzed respondents’ adjacent open-ended written answers to uncover more detailed information about respondents’ postpartum contraception behavior and interpretations of the survey question.

-Through systematic recategorization, we found that qualitative responses added to and altered quantitative contraception behavior reports, and that reasons for contraception non-use are incompletely captured by existing response categories.

What do these findings mean?

-These findings suggest that researchers may be able to gain a better understanding of postpartum pregnancy prevention by using mixed-method analyses on categorical and open-ended responses. 

-While qualitative inquiry may yield subjective interpretation, rigorous pragmatic data management can strengthen survey questions through open-text responses not captured in existing quantitative data.

-Family planning professionals may be able to obtain significant and detailed information about contraceptive behavior from open-ended responses, and this information may be useful to adjust clinical approaches to postpartum contraception conversations.

2. Methods: Line 148: Please change this to “mixed-method” lens. 

3. Discussion: Lines 254-258: Please revise to: “The statistically significant differences suggest the importance of re-examining survey responses to improve data quality; however, it should be noted that the original and recoded findings yield similar measures of contraception use. Overall, these PRAMS questions illuminated an unclear understanding of the definition of pregnancy prevention in some instances, and this may have contributed to inconsistent answers based on respondent interpretation.”

4. Discussion: Line 266: Please remove the word “rich” from the sentence. 

5. Discussion: At line 268, and line 272, please avoid the use of italics for emphasis.

6. Table 1: Please report p values as p<0.001 for each comparison where applicable, rather than p=0.000. Please indicate if the comparison of original and recoded responses for who report “No” to the question on use of contraception method was tested (it is not indicated as p<0.001).

7. Figure 1 and Figure 2: Thank you for providing the legends for the figures. Please note that the figures themselves are no longer included with this version of the submission. These were previously included as files named “Survey Figure 1” and “Survey Figure 2”. Please provide and re-incorporate these into the manuscript.

PRESS

Sincerely, 

Caitlin Moyer, Ph.D. 

Associate Editor 

PLOS Medicine